# Thruster Fault Diagnostics and Fault Tolerant Control for Autonomous Underwater Vehicle with Ocean Currents

**Qunhong Tian** [1,*]**, Tao Wang** [1]**, Bing Liu** [1] **and Guangtao Ran** [2,3]

1    College of Mechanical and Electronic Engineering, Shandong University of Science and Technology, Qingdao 266590, China; wangt@sdust.edu.cn (T.W.); metrc@sdust.edu.cn (B.L.)
2    Department of Control Science and Engineering, Harbin Institute of Technology, Harbin 150001, China; ranguangtao@hit.edu.cn
3    Department of Electrical and Computer Engineering, University of Alberta, Edmonton, AB T6G 1H9, Canada
*    Correspondence: tianqunhong@sdust.edu.cn

**Abstract:** Autonomous underwater vehicle (AUV) is one of the most important exploration tools in the ocean underwater environment, whose movement is realized by the underwater thrusters, however, the thruster fault happens frequently in engineering practice. Ocean currents perturbations could produce noise for thruster fault diagnosis, in order to solve the thruster fault diagnostics, a possibilistic fuzzy C-means (PFCM) algorithm is proposed to realize the fault classification in this paper. On the basis of the results of fault diagnostics, a fuzzy control strategy is proposed to solve the fault tolerant control for AUV. Considering the uncertainty of ocean currents, it proposes a min-max robust optimization problem to optimize the fuzzy controller, which is solved by a cooperative particle swarm optimization (CPSO) algorithm. Simulation and underwater experiments are used to verify the accuracy and feasibility of the proposed method of thruster fault diagnostics and fault tolerant control.

**Keywords:** autonomous underwater vehicle; thruster fault diagnostics; fault tolerant control; robust optimization; ocean currents

## 1. Introduction

An autonomous underwater vehicle (AUV) is one of the most important exploration tools in the ocean underwater environment. As an important part of AUV, the thruster directly determines the efficiency and safety with strong working intensity for AUV, However, the thruster fault usually happens in engineering practice [1,2]. Therefore, how to make thruster fault diagnosis and fault tolerant control for AUV is the premise for completing underwater missions [3,4].

There have been many works applied to AUV fault diagnosis. A Gaussian particle filtering algorithm is presented to estimate the AUV failure model, the Bayes algorithm is used to realize the AUV thruster fault detection [5]. For solving the fault diagnosis of AUV actuators, a diagnostic network is proposed based on extreme learning and a wide convolutional neural network [6]. Through experimental data analysis, a feature calculation method is presented to solve the weak faults thruster faults, which provides accurate and concise information for fault severity identification [7]. A fault diagnosis method is presented based on deep learning and attention mechanism for AUV, a data attention mechanism is developed for realizing dynamic decorrelation, multi-layer perceptron is used for fault detection [8]. From training datasets gathered in previous AUV operations directly, the Bayesian nonparametric technique is used for modelling the vehicle's performance including faults, in the light of the Kullback-Leibler divergence measure, a nearest-neighbor classifier is used to accomplish the fault diagnosis [9]. In summary, the above studies have given some methods to solve the AUV fault diagnosis. However, ocean currents perturbations could produce noise for thruster fault diagnosis, the above methods are

difficult to be used for AUV fault diagnosis with ocean currents in practice effectively. The above methods also do not consider how to control AUV to complete the underwater missions with minor faults.

Fault tolerant control is the technology to ensure the AUV for completing the underwater mission with faults [10,11]. In order to realize the fault tolerant control, it develops a model-parameter-free control strategy for AUV trajectory tracking, tracking controller is designed through the employment of sliding mode control technology without utilizing model parameters. However, the sliding mode control easily lead to the chattering of the AUV control system [12]. In order to solve the problem of thruster fault tolerant control for AUV, a fault tolerant control method is proposed in the light of the sliding mode theory, the adaptive law is developed for the proposed controller to mitigate the chattering phenomenon [13]. In order to further improve the performance of the fault tolerant control, some intelligent methods are investigated [4,14,15]. An iterative learning algorithm is proposed to process the propeller failure for AUV based on an extended state observer, a fuzzy logic controller is introduced to deal with the fuzzification of the parameters of a saturated proportional-derivative controller and extended state observer [14]. Combined with the backstepping method, a single critic network based on adaptive dynamic programming is used to deal with the AUV fault tolerant control. It designs an online policy iteration algorithm in light of the estimated system states [4]. To further conduct the effect of the ocean currents, the fault tolerant issue is transformed into an optimal control problem by the adaptive dynamic programming method, the neural-network estimator is developed to estimate ocean currents [15], however, it is difficult to establish the ocean current accurately in practice. In summary, although the above research has given some methods for fault tolerant control for AUV, they are difficult to be used in an environment with ocean currents.

Ocean currents perturbations could produce noise for thruster fault diagnosis. In this paper, in order to solve the problem of the thruster fault diagnostics and fault tolerant control for AUV with ocean currents, the possibilistic fuzzy C-means (PFCM) algorithm is proposed for realizing the thruster fault diagnostics effectively. Once the thruster fault is diagnosed, based on the fault diagnosis results, a fault tolerant control is presented by the fuzzy controller, to improve the performance of the fuzzy controller, a robust optimization problem is proposed by considering the uncertainty of ocean currents, which is solved by the proposed co-evolutionary (CPSO) algorithm, finally, it forms a mechanism of diagnostics and control strategy to accomplish the missions.

The rest of this paper is given as follows. Section 2 presents the AUV mathematical models; Section 3 gives the algorithm for AUV fault diagnostics and fault tolerant control; Section 4 discusses the effectiveness of the proposed method based on different scenarios; Section 5 concludes the paper.

## 2. Mathematical Models of AUV

In this section, the problem description is given for the AUV firstly, and then the AUV models are discussed.

### 2.1. Problem Description

AUV works in a complex marine environment, which is a complex dynamic system with strong nonlinearity. Due to the complexity and unpredictability of the marine environment, thrusters are easy to fail. However, when the thruster fails, the expected task of the AUV cannot be completed, or AUV may be destroyed directly, which will cause extremely serious losses and may pollute the environment.

Thruster fault diagnostics is the premise to solve the above problems, which include the type of motor fault, propeller enwinding by foreign matter, propeller blade damage, thruster idling, and so on. However, AUV is greatly affected by the external disturbance of ocean currents, the external interference and fault are difficult to be separated, which takes great difficulty for AUV fault diagnosis. Meanwhile, the ocean currents increase the

difficulty of controlling the AUV to accomplish the missions with thruster fault. Figure 1 gives the design process of thruster fault diagnostics and faults tolerant control for AUV, the PFCM algorithm is proposed to realize the thruster fault diagnostics. Once the thruster fault is diagnosed, based on the fault diagnosis results, a fault tolerant control is presented by the fuzzy controller, in order to improve the performance of the fuzzy controller, a robust optimization problem is proposed by considering the uncertainty of ocean currents, which is solved by the proposed CPSO algorithm, finally, it forms a mechanism of fault diagnostics and tolerant control strategy to accomplish the missions.

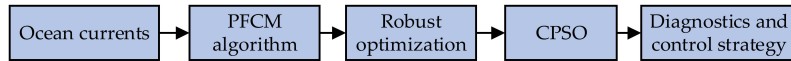

**Figure 1.** Design process of thruster fault diagnostics and fault tolerant control for AUV.

*2.2. AUV KINEMATIC model*

Figure 2 gives two coordinate systems for AUV to obtain the kinematic model, one is the earth-frame $\{O - X, Y, Z\}$, the other is the body-fixed frame $\{O_1 - X_1, Y_1, Z_1\}$. The AUV kinematic model can deal with the geometric aspects of motion, which is written in a general form as follows [16–19]:

$$\dot{\eta} = \begin{bmatrix} J_1 & \mathbf{0}_{3\times3} \\ \mathbf{0}_{3\times3} & J_2 \end{bmatrix} v \tag{1}$$

$$\begin{bmatrix} \dot{x} & \dot{y} & \dot{z} & \dot{\phi} & \dot{\theta} & \dot{\psi} \end{bmatrix}^T = \begin{bmatrix} J_1 & \mathbf{0}_{3\times3} \\ \mathbf{0}_{3\times3} & J_2 \end{bmatrix} \begin{bmatrix} u & v & w & p & q & r \end{bmatrix}^T \tag{2}$$

where the vector $\eta = \begin{bmatrix} x & y & z & \phi & \theta & \psi \end{bmatrix}^T$ denotes the position and orientation of AUV in the Earth-frame, $x, y, z$ represent the position, $\phi, \theta, \psi$ are the Euler angles of roll, pitch and yaw angles respectively; $v = \begin{bmatrix} u & v & w & p & q & r \end{bmatrix}$ denotes the translational and rotational velocities in the body-fixed frame, $u, v, w$ are the surge, sway and heave components respectively, $p, q, r$ are the roll, pitch and yaw rates respectively; $J_1$ and $J_2$ are the coordinate transformation matrixes, which are given as follows:

$$J_1 = \begin{bmatrix} \cos\theta\cos\psi & \sin\theta\sin\phi\cos\psi - \cos\phi\sin\psi & \sin\theta\sin\psi + \sin\theta\cos\phi\cos\psi \\ \cos\theta\sin\psi & \sin\theta\sin\phi\sin\psi + \cos\phi\cos\psi & \sin\theta\cos\phi\sin\psi - \sin\phi\cos\psi \\ -\sin\theta & \sin\phi\cos\theta & \cos\phi\cos\theta \end{bmatrix} \tag{3}$$

$$J_2 = \begin{bmatrix} 1 & \sin\phi\tan\theta & \cos\phi\tan\theta \\ 0 & \cos\phi & -\sin\phi \\ 0 & \sin\phi/\cos\theta & \cos\phi/\cos\theta \end{bmatrix} \tag{4}$$

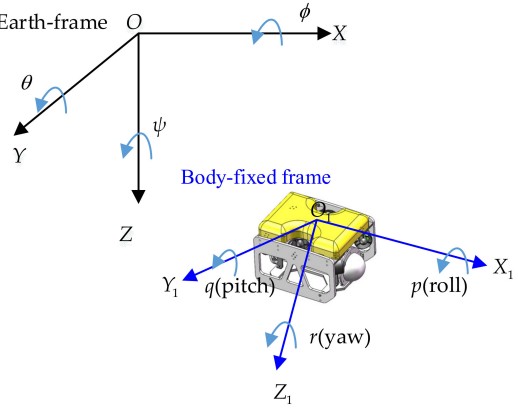

**Figure 2.** Coordinate systems for AUV.

### 2.3. AUV Dynamic Model

It can describe the general motion of AUV with six degrees of freedom dynamic equation as follows [17,18,20]:

$$\boldsymbol{M\dot{v}} + \boldsymbol{C(v)V} + \boldsymbol{D(v)v} + \boldsymbol{g(\eta)} = \boldsymbol{\tau} \tag{5}$$

where $\boldsymbol{M} \in \boldsymbol{R}^{6\times6}$ is the inertial matrix; $\boldsymbol{v}$ is the position and orientation vector; $\boldsymbol{C} \in R^{6\times6}$ is the matrix of Coriolis and Centripetal terms; $\boldsymbol{g} \in \boldsymbol{R}^{6\times6}$ is the gravitational terms matrix; $\boldsymbol{D} \in \boldsymbol{R}^{6\times6}$ is the damping matrix; $\boldsymbol{\tau}$ is the control forces vector. Figure 3 shows the planform of the designed AUV, it assumes that the center of gravity is at the same point as the center of buoyancy for AUV, the translational motion and rotational motion are expressed by six equations as follows:

$$\begin{aligned} & m\left(\dot{u} - vr + wq - x_g\left(q^2 + r^2\right) + z_g(pr + q)\right) \\ & = X_{HS} + X_u|u| + X_{\dot{u}}u + X_{wq}wq + X_{qq}qq + X_{vr}vr + X_{rr}rr + X_{prop} \end{aligned} \tag{6}$$

$$\begin{aligned} & m\left(\dot{v} - wp + ur - z_g\left(qr - \dot{p}\right) + x_g\left(pq + \dot{r}\right)\right) \\ & = Y_{HS} + Y_{v|v|}v|v| + Y_{r|r|}r|r| + Y_{\dot{v}}\dot{v} + Y_{\dot{r}}\dot{r} + Y_{ur}ur + Y_{wp}wp + Y_{pq}pq + Y_{uv}uv + Y_{uu\delta_r}u^2\delta_r \end{aligned} \tag{7}$$

$$\begin{aligned} & m\left(\dot{w} - uq + vp - z_g\left(q^2 + p^2\right) + x_g\left(rp + \dot{q}\right)\right) \\ & = Z_{HS} + Z_{w|w|}w|w| + Z_{q|q|}q|q| + Z_{\dot{w}}\dot{w} + Z_{\dot{q}}\dot{q} + Z_{uq}uq + Z_{vp}vp + Z_{pr}pr + Z_{uw}uw + Z_{uu\delta_s}u^2\delta_s \end{aligned} \tag{8}$$

$$\begin{aligned} & I_{xx}\dot{p} + \left(I_{zz} - I_{yy}\right)qr + m\left|-z_g\left(\dot{v} - wp + ur\right)\right| \\ & = K_{HS} + K_{p|p|}p|p| + k_{p|p|}p|p| + k_{\dot{p}}\dot{p} + k_{prop} \end{aligned} \tag{9}$$

$$\begin{aligned} & I_{yy}\dot{q} + \left(I_{xx} - I_{zz}\right)pr + m\left|z_g\left(\dot{u} - vr + wq\right) - x_g(\dot{w} - uq + vp)\right| = \\ & M_{HS} + M_{w|w|}q|q| + M_{q|q|}q|q| + M_{\dot{w}}\dot{w} + M_{\dot{q}}\dot{q} + M_{uq}uq + M_{vp}vp + M_{rp}rp + M_{uw}uw + M_{uu\delta_s}u^2\delta_s \end{aligned} \tag{10}$$

$$\begin{aligned} & I_{zz}\dot{r} + \left(I_{yy} - I_{xx}\right)qp + m\left|x_g\left(\dot{v} - wp + ur\right)\right| = \\ & N_{HS} + N_{v|v|}v|v| + N_{r|r|}r|r| + N_{\dot{v}}\dot{v} + N_{\dot{r}}\dot{r} + N_{ur}ur + N_{wp}wp + N_{pq}pq + N_{uv}uv + N_{uu\delta_r}u^2\delta_r \end{aligned} \tag{11}$$

$$F = \begin{bmatrix} 0 & 0 & \cos\beta & \cos\beta & -\cos\beta & -\cos\beta \\ 0 & 0 & \sin\beta & -\sin\beta & \sin\beta & -\sin\beta \\ 0 & 1 & 0 & 0 & 0 & 0 \\ b_v & b_v & \sin\beta \cdot c_h & \sin\beta \cdot c_h & -\sin\beta \cdot c_h & \sin\beta \cdot c_h \\ a_v & a_v & \cos\beta \cdot c_h & \cos\beta \cdot c_h & -\cos\beta \cdot c_h & -\cos\beta \cdot c_h \\ 0 & 0 & B_1 & B_2 & B_3 & B_4 \end{bmatrix} \begin{bmatrix} F_1 \\ F_2 \\ F_3 \\ F_4 \\ F_5 \\ F_6 \end{bmatrix} \tag{12}$$

where $B_1 = \cos\beta \cdot b_h + \sin\beta \cdot a_h$, $B_2 = -\cos\beta \cdot b_h - \sin\beta \cdot a_h$, $B_3 = -\cos\beta \cdot b_h - \sin\beta \cdot a_h$, $B_4 = \cos\beta \cdot b_h - \sin\beta \cdot a_h$. $F_1$ is the thrust of the left vertical thruster; $F_2$ is the thrust of the right vertical thruster. $F_3$ is the thrust of the left front horizontal thruster; $F_4$ is the thrust of the right front horizontal thruster; $F_5$ is the thrust of the left rear horizontal thruster thrust; $F_6$ is the thrust of the right rear horizontal thruster thrust. $a_v$ is the distance between the center of vertical thruster and $X_1O_1Z_1$ plane; $b_v$ is the distance between the center of vertical thruster and $Y_1O_1Z_1$ plane; $a_h$ is the distance between the center of horizontal thruster and $X_1O_1Z_1$ plane; $b_h$ is the distance between the center of horizontal thruster and $Y_1O_1Z_1$ plane; $c_h$ is the distance between the center of horizontal thruster and $X_1O_1Y_1$ plane.

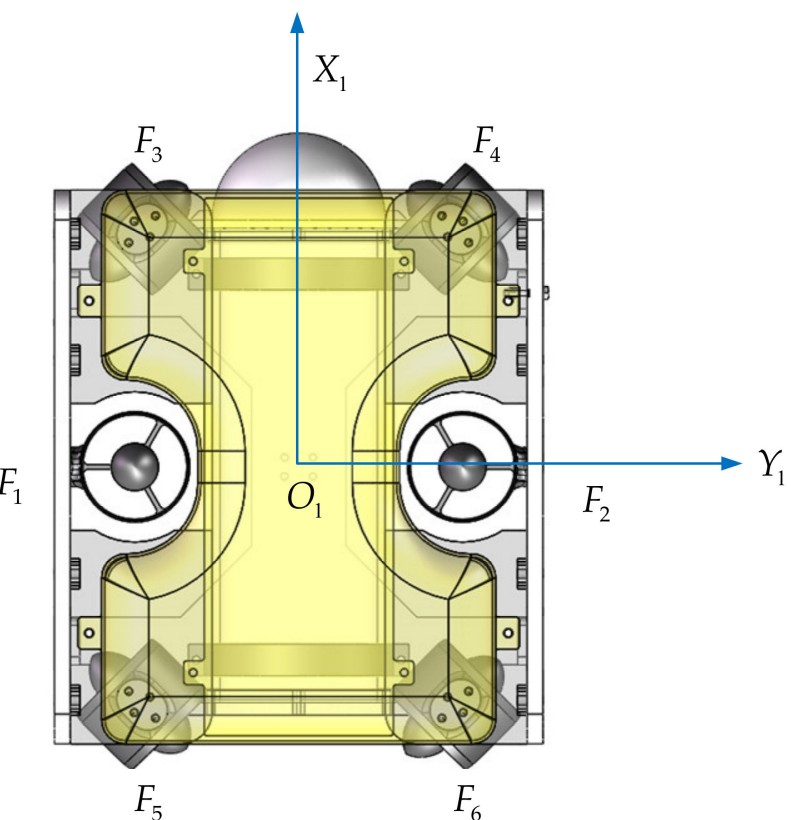

**Figure 3.** Planform of the designed AUV.

### 3. Thruster Fault Diagnostics and Fault Tolerant Control

Thruster fault diagnostics and fault tolerant control problem is discussed in this section. Based on the results of thruster fault diagnostics by PFCM algorithm, a fuzzy controller is proposed for AUV fault tolerant control, which is optimized with the proposed CPSO algorithm, the control performance is ensured with the robust optimization design.

### 3.1. Thruster Fault Diagnostics for AUV

AUV thruster is usually constituted by a motor, reducer, propeller, controller, and so on. The battery pack provides the drive energy, motor, and reducer as the actuator, the output voltage is controlled by the received upper computer instruction by the driver controller, further to control motor speed, motor and speed reducer drive screw rotation. Generally, the speed is proportional to the thrust. The propeller provides the thrust to realize the variable speed sailing of AUV, the drive controller uploads the real-time operation parameters of the propeller to the upper computer. The propeller protection cover is used to avoid the propeller damage caused by the impact of fish or other objects. In this paper, the type of the thruster fault is given as follows: motor fault, propeller enwinding by foreign matter, propeller blade damage, thruster got stuck, thruster idling. The voltage, current, and speed of the thruster are used to judge whether the thruster is faulty.

It is well known that it is difficult for fault diagnosis of nonlinear systems [21,22]. Moreover, ocean currents perturbations could produce noise and further increase the difficulty of thruster fault diagnosis. To solve the thruster diagnosis of AUV nonlinear system with ocean currents, it proposes the PFCM algorithm. PFCM algorithm is one popular clustering method, it is highly sensitive to noise and outliers, and the size of the clusters [23,24]. The algorithm is an unsupervised technique, the data is clustered based on similarities and dissimilarities, which are measured via distances of the cluster centers to the data points. The clustering results are described by introducing membership and probability partition matrixes.

PFCM algorithm is proposed with optimization as follows [25].

$$\min \quad J_M(U, T, V) = \sum_{i=1}^{n} \sum_{j=1}^{c} (au_{ij}^m + bt_{ij}^p)d_{ij}^2 + \sum_{i=1}^{n} \eta_i \sum_{j=1}^{c} (1 - t_{ij})^p \tag{13}$$

$$s.t. \quad \sum_{i=1}^{c} u_{ij} = 1$$

$$\eta_i = K \sum_{j=1}^{n} u_{ij}^m d_{ij}^2 / \sum_{j=1}^{n} u_{ij}^m \tag{14}$$

where $c$ is the number of clusters; $m > 1$ is the degree of fuzziness; $n$ is the number of data points; $a$ and $b$ are the constants ($a > 0$, $b > 0$), which represent relative importance of fuzzy and possibilistic terms respectively, the larger value of $b$, the better the ability to resist noise points; $U = [u_{ij}]_{c \times n}$ is the membership degrees matrix ($0 \leq u_{ij}$); $T = [t_{ij}]_{c \times n}$ is the typicality matrix ($t_{ij} \leq 1$); $V = [v_{ij}]_{c \times n}$ is the cluster centers matrix; $p > 1$ is the possibilistic exponent, $d_{ij}$ is the distance between the cluster center ($v_i$) and data point ($x_i$); $\eta_i$ is the penalty factor, $K$ is a constant.

The objective function in Equation (13) can be solved via an iterative procedure as follows:

$$u_{ij} = \frac{1}{\sum_{k=1}^{c} \left( \frac{d_{ij}}{d_{kj}} \right)^{2/m-1}} \tag{15}$$

$$t_{ij} = \frac{1}{1 + \left( \frac{b}{\eta_i} d_{ij}^2 \right)^{\frac{1}{p-1}}} \tag{16}$$

$$v_i = \frac{\sum_{j=1}^{n} \left( au_{ij}^m + bt_{ij}^p \right) x_j}{\sum_{j=1}^{n} \left( au_{ij}^m + bt_{ij}^p \right)} \tag{17}$$

according to the above Equations (15)–(17), it can obtain the optimal degree of membership and cluster center.

### 3.2. Fault Tolerant Control for AUV

Fuzzy theory can describe the uncertainty of the system, it has been used to solve the problem of fault diagnosis and control effectively [26–28]. Therefore, in this paper, fuzzy controller is proposed to solve the fault tolerant control problem for AUV path tracking, the fuzzy control includes the fuzzification, fuzzy inference and defuzzification. For the fuzzification operation, the Gaussian function is selected as the membership function of fuzzy variable; Table 1 gives the fuzzy control rule for fuzzy inference; centroid method is used to realize the defuzzification operation. The input parameters of the fuzzy controller are position error $e(t)$ and its derivative $\dot{e}(t)$, the output parameters of the fuzzy controller are angles and those derivatives. Figures 4 and 5 give the membership functions for position error and its derivative respectively. $\lambda_1, \lambda_2, \lambda_3, \lambda_4, \lambda_5, \lambda_6, \lambda_7$ are the mean of the normal distribution for Gaussian membership function of position error. It includes seven fuzzy states: $NB(\lambda_1), NM(\lambda_2), NS(\lambda_3), ZO(\lambda_4), PS(\lambda_5), PM(\lambda_6), PB(\lambda_7)$. $\beta_1, \beta_2, \beta_3, \beta_4, \beta_5, \beta_6, \beta_7$ are the mean of the normal distribution for Gaussian membership function of the derivative of position error. It includes seven fuzzy states: $NB(\beta_1), NM(\beta_2), NS(\beta_3), ZO(\beta_4), PS(\beta_5), PM(\beta_6), PB(\beta_7)$.

**Table 1.** Fuzzy control rules for AUV.

| $e(t)/\dot{e}(t)$ | **NB** | **NM** | **NS** | **ZO** | **PS** | **PM** | **PB** |
|---|---|---|---|---|---|---|---|
| NB | NB | NB | NM | NS | NS | ZO | PM |
| NM | NB | NM | NS | ZO | ZO | PS | PM |
| NS | NB | NM | NS | ZO | PS | PS | PM |
| ZO | NB | NM | NS | ZO | PS | PM | PB |
| PS | NM | NS | NS | ZO | PS | PM | PB |
| PM | NM | NS | ZO | ZO | PS | PM | PB |
| PB | NM | ZO | PS | PS | PM | PB | PB |

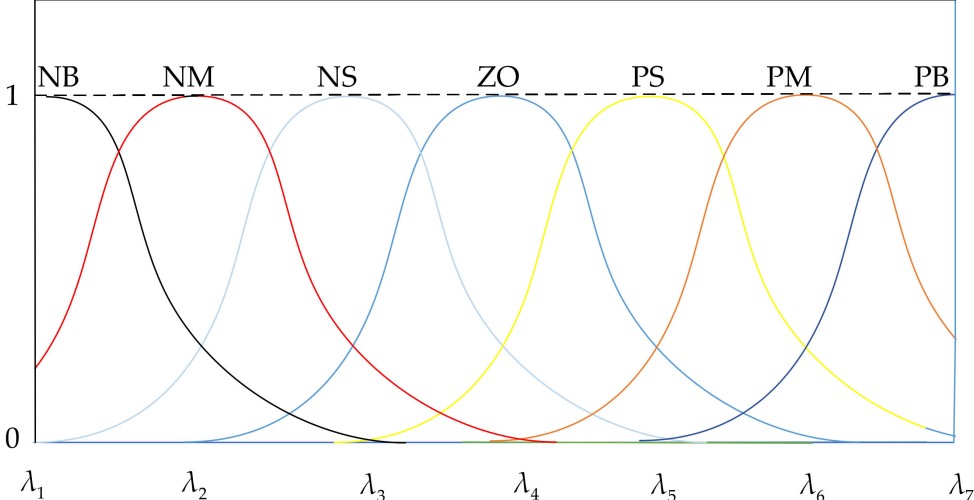

**Figure 4.** Membership function for position error.

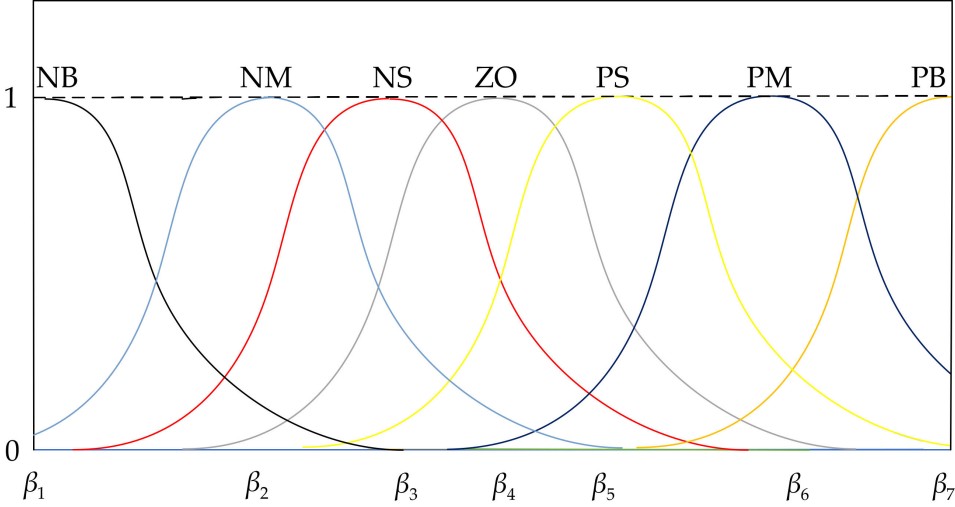

**Figure 5.** Membership function for the derivative of position error.

### 3.3. Robust Optimization for AUV

Tracking error between the desired and tracking path is the important performance index for path tracking results, which is replaced by the average of the total absolute of the position errors in this paper, it can be given as follows:

$$f_E = \frac{1}{T}\sum_{t=0}^{T}|e_t| \tag{18}$$

where $e_t$ is one position error at the point $q_t$ of the tracking path; $|e_t|$ is the absolute value of $e_t$, $f_E$ is the average of the total $|e_t|$, $T$ is the total number of points of AUV tracking path.

It defines that the start and target points are $(x_0, \ y_0, \ z_0)$ and $(x_T, \ y_T, \ z_T)$ for the tracking path respectively, the tracking path can consist of a sequence of points $A = [(x_0, \ y_0, \ z_0), \cdots, (x_t, \ y_t, \ z_t), \cdots, (x_T, \ y_T, \ z_T)]$.

$$
\begin{aligned}
\min \quad & f_E(A) \\
s.t. \quad & x_{\min} \leq x_t \leq x_{\max} \\
& y_{\min} \leq y_t \leq y_{\max} \\
& z_{\min} \leq z_t \leq z_{\max} \\
& \dot{\boldsymbol{\eta}} = \begin{bmatrix} J_1 & \mathbf{0}_{3\times3} \\ \mathbf{0}_{3\times3} & J_2 \end{bmatrix} V
\end{aligned}
\tag{19}
$$

where $x_m, y_m, z_m$ are the decision variables of the optimization problem for AUV path tracking; $(x_{\max}, y_{\max}, z_{\max})$ and $(x_{\min}, y_{\min}, z_{\min})$ are the maximum and minimum coordinate position points, respectively.

Considering the uncertain ocean currents, Equation (19) can be transformed into a robust optimization problem as follows:

$$
\begin{aligned}
\min_{(x_m,y_m,z_m)} \ \max_{(u_c,v_c,w_c)} \quad & f(x_m, y_m, z_m, u_c, v_c, w_c) \\
s.t. \quad & x_{\min} \leq x_m \leq x_{\max} \\
& y_{\min} \leq y_m \leq y_{\max} \\
& z_{\min} \leq z_m \leq z_{\max} \\
& \dot{\boldsymbol{\eta}} = \begin{bmatrix} J_1 & \mathbf{0}_{3\times3} \\ \mathbf{0}_{3\times3} & J_2 \end{bmatrix} V \\
& u_{\mathrm{minc}} \leq u_c \leq u_{\mathrm{maxc}} \\
& v_{\mathrm{minc}} \leq v_c \leq v_{\mathrm{maxc}} \\
& w_{\mathrm{minc}} \leq w_c \leq w_{\mathrm{maxc}}
\end{aligned}
\tag{20}
$$

where $\begin{bmatrix} u_{\mathrm{minc}} & v_{\mathrm{minc}} & w_{\mathrm{minc}} \end{bmatrix}$ and $\begin{bmatrix} u_{\mathrm{maxc}} & v_{\mathrm{maxc}} & w_{\mathrm{maxc}} \end{bmatrix}$ are the minimum and maximum values of the components of ocean currents. Equation (20) is the robust optimization problem, which is also called "min-max" optimization problem for the AUV path tracking, whose goal is to find the robust solution for the tracking path with the best performance in all the worst ocean currents.

CPSO is proposed to solve the robust optimization problem (20), which can find a good solution to the "min-max" optimization problem for the AUV path tracking. The CPSO algorithm involves two populations $P_1$ and $P_2$, each population evolves independently and tied together via the fitness evaluation [29,30]. The first population $P_1$ is used to evolve the decision variables $(x_m, y_m, z_m)$, the second population $P_2$ is used to evolve the ocean currents $(u_c, v_c, w_c)$.

For the first population, the fitness function of decision variables is given by

$$
G(x_m, y_m, z_m) = \max_{u_c, v_c, w_c \in P_2} f(x_m, y_m, z_m, u_c, v_c, w_c)
\tag{21}
$$

which is to be minimized.

For the second population, the fitness function of ocean currents is given by

$$H(u_c, v_c, w_c) = \min_{x_m, y_m, z_m \in P_1} f(x_m, y_m, z_m, u_c, v_c, w_c) \tag{22}$$

which is to be maximized.

### 3.4. Thruster Fault Diagnostics and Fault Tolerant Control Algorithm

For the CPSO algorithm, in the light of Equation (21), the global best value in $P_1$ is gotten as the solution. Based on Equation (22), the globally best values in $P_2$ are obtained as the scenarios for ocean currents. According to the above design principles, the optimal tracking path can be obtained.

Figure 6 gives the flowchart for the thruster fault diagnostics and fault tolerant control of AUV, the corresponding steps are given in detail in Algorithm 1.

---

**Algorithm 1:** Thruster fault diagnostics and fault tolerant control for AUV

---

1: Initializing the parameters $m$, $p$, $U$, $T$, $V$ and so on for thruster fault diagnostics;

2: Calculating the penalty factor $\eta_i$ based on Equation (14), updating $U$, $T$, $V$ based on Equations (15)–(17) respectively;

3: If the iterations $N_d$ are smaller than the given maximum number of times ($N_{d\_max}$), obtaining the final $U$, $T$, $V$; else if go to Step 2.

4: Considering the effects of ocean currents, establishing the robust optimization model for AUV fault tolerant control systems.

5: Establishing the models of the evaluation functions (21) and (22) for $P_1$ and $P_2$ respectively.

6: Initializing the two populations randomly, evaluating each population co-evolutionarily by using (21) and (22), respectively.

7: Evolving the population $P_1$ based on (21); replacing the global best ($g_{best}$) and personal best ($p_{best}$) particle positions.

8: If the iterations ($N_{i\_1}$) is smaller than the given maximum number of times ($N_{m\_1}$), go to the next step, else if go to Step 7.

9: Evolving the population $P_2$ based on (22); replacing $g_{best}$ and $p_{best}$ particle positions.

10: If the iterations $N_{i\_2}$ are smaller than the given maximum number of times ($N_{m\_2}$), go to the next step, else if, go to step 9.

11: If the iterations $N_{i\_3}$ are smaller than the given maximum number of times ($N_{m\_3}$), obtaining the optimal parameters of the membership function, then getting the final tracking points, end the program; else if go to Step 6.

---

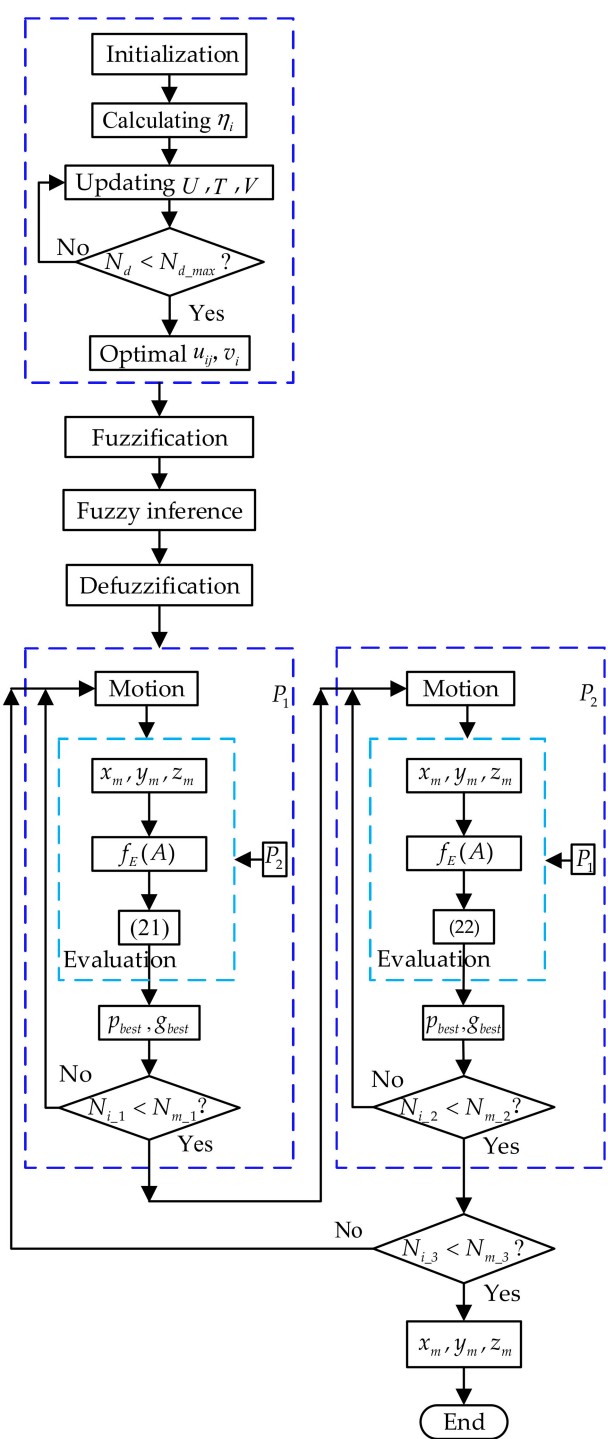

**Figure 6.** Flowchart for thruster fault diagnostics and fault tolerant control for AUV.

## 4. Simulation and Experiment Analysis

In this paper, different experiments are given to analyze the performance of the proposed thruster fault diagnostics and fault tolerant control method for AUV, Figure 7 shows the designed AUV, and Figure 8 gives the AUV thruster in practice. Based on the test data of the thruster fault, the results of the fault type are obtained for AUV thruster fault diagnostics. AUV is equipped with six underwater propellers, among which the propeller at the top of the AUV is used to control the sinking and floating of the AUV, and the other four propellers at the front and back are used to control the forward, backward, and steering of the AUV. The speed of AUV is set as 0.15 m/s, and the speed of uncertain ocean

current is set as 0–0.08 m/s, whose direction is random. Sine path, circular path, rectangular path, and irregularity path are given to illustrate the tracking effect with thruster fault. The algorithm is coded in MATLAB R2019a and simulations are run on the PC with 2.00 GHz CPU/8 GB RAM.

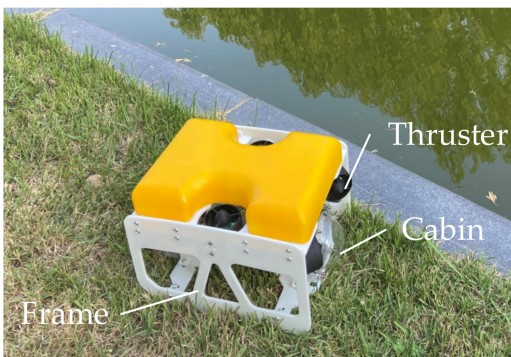

**Figure 7.** The designed AUV.

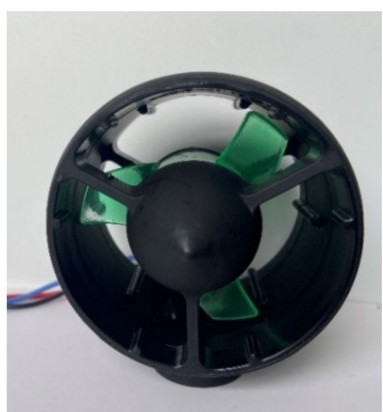

**Figure 8.** The thruster of AUV.

In order to test the effectiveness of the proposed thruster fault diagnostics, a data set with 300 groups is obtained from the underwater experiment for our designed AUV. Each group data is composed of voltage, current, and speed of the thrusters, which can denote the characteristic of six types of thruster operation: motor fault, propeller enwinding by foreign matter, propeller blade damage, thruster got stuck, thruster idling, and normal operation. Therefore, it assumes that the number of the clustering centers is 6. Figure 9 shows the classification results by the proposed PFCM algorithm. The center of clustering of the thruster stuck state is $(11.98 \text{ V}, 0.58 \text{ A}, 0.75 \text{ r/s})$, the center of clustering of the propeller enwinding state is $(12.2 \text{ V}, 0.41 \text{ A}, 12.07 \text{ r/s})$, the center of clustering of the thruster normal operation is $(12.48 \text{ V}, 0.35 \text{ A}, 16.9 \text{ r/s})$, the center of clustering of the propeller damage is $(12.78 \text{ V}, 0.27 \text{ A}, 21.01 \text{ r/s})$, the center of clustering of the thruster idling state is $(13 \text{ V}, 0.17 \text{ A}, 25 \text{ r/s})$, the center of clustering of the motor fault is $(13 \text{ V}, 0.05 \text{ A}, 0.77 \text{ r/s})$. The signals of six different fault types are closely clustered around their respective clustering centers after classification by the PFCM algorithm. The proposed fault detection algorithm can accurately identify the fault types of AUV and effectively classify them.

Because the thrusters of AUV are often immersed in seawater, the probability of failure is significantly improved after the corrosion of seawater. AUV operation in the ocean may be enwound by marine plants or marine organisms, which affects the performance of the thrusters. The thruster is the main forward power of AUV, if the above phenomenon occurs, it affects the velocity, heading angle, and the safety of AUV, and even leads to the AUV being unrecoverable. Therefore, in this paper, after detecting the fault type of AUV based on the PFCM algorithm, the corresponding fault tolerant control is adopted according to the

fault type and degree. By reducing the thrust of other thrusters, the AUV can continue to move to accomplish the missions. If the AUV loses all power, it can be stopped and floated up for recovery. Therefore, the fault tolerant control proposed in this paper only applies to the fault types of AUV power loss caused by AUV enwinding or propeller damage.

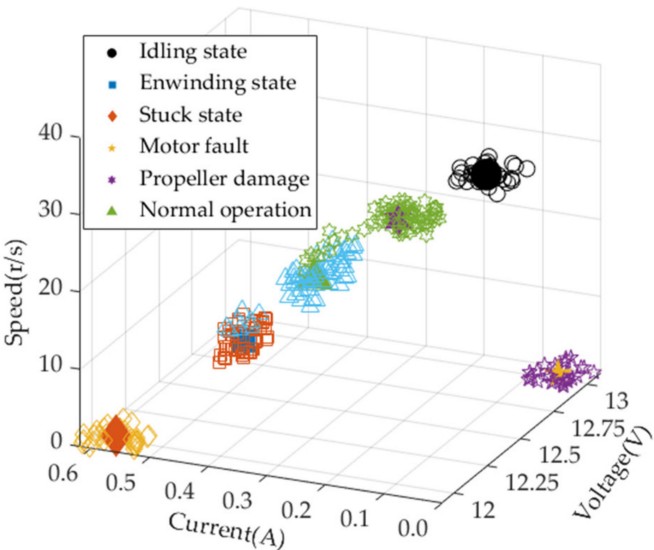

**Figure 9.** The classification results by the proposed PFCM algorithm.

When a thruster fault happens, it loses thrust and its torque balance is broken, which leads the change of heading angle and takes off its desired path. For example, if the left thruster ($F_5$) of the AUV fault happens and its thrust decreases, the thrust of its adjacent thruster ($F_6$) should be correspondingly reduced to balance the torque of the AUV. If the two adjacent thruster faults happen, the thrust of the thruster with a larger thrust is reduced accordingly to make its torque reach balance. If these two thrusters ($F_5$, $F_6$) have a large degree of fault, the thrusters ($F_3$ and $F_4$) are responsible for AUV regression on the opposite side, which can complete the follow-up tasks or realize turning back.

In the complex environment, there are many factors that affect the AUV operation, such as obstacles, ocean currents, and fish schools. Therefore, the AUV path is not a line, the curved path is an essential to the AUV path. This paper gives a sine curve path as follows:

$$
\begin{cases}
x(t_1) = t_1 \\
y(t_1) = 80\sin(t_1\pi/80) \\
z(t_1) = 0
\end{cases}
\tag{23}
$$

where $t_1 \in [0:0.25:360]$, it assumes that the sine curve is constituted of 1440 points $[(0, 0, 0), (0.25, 0.78, 0), \cdots, (360, 0, 0)]$, the tracking start point is $(0, 0, 0)$. The initial position, angle, initial velocity, and expected velocity are set as $(x, y, z) = (0, 0, 0)$ m,$(\varphi, \theta, \phi) = (0°, 0°, 0°)$, $(u, v, w) = (0, 0, 0)$ [kn], $(u, v, w) = (0.2, 0, 0)$ [kn] for AUV. If the thruster is enwound with foreign matter. Based on the results of thruster fault diagnostics, Figure 10 gives the tracking results for the sine path by the proposed fault tolerant control algorithm, the tracking path length is 832.91 m. Figure 11 shows the position error for the corresponding sine path tracking, the average position errors is 0.27 m, the range of the tracking error is $[-1.44\,\text{m}, 1.91\,\text{m}]$, and the standard deviation of the tracking error is 0.16 m. The proposed algorithm can realize the sine path tracking with thruster enwinding effectively.

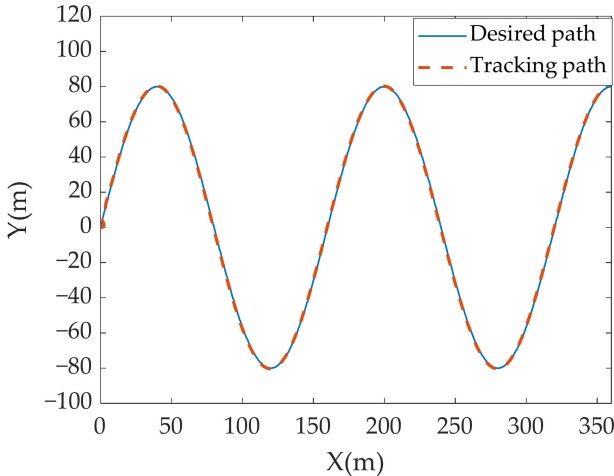

**Figure 10.** Tracking for sine path.

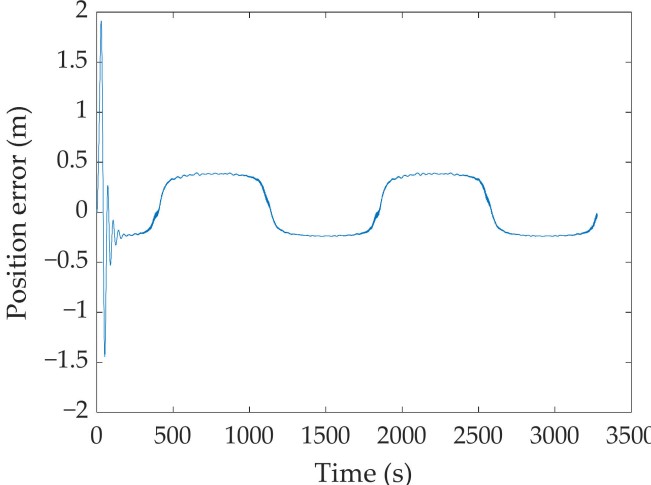

**Figure 11.** Position error for the sine path tracking.

In the practice, in order to complete some specific tasks, AUV needs to move around the detected object in a circle. Therefore, the circular curve is also one of the key paths of AUV path tracking. The circular path is described as follows:

$$\begin{cases} x(t_2) = 55 + 40\cos(t_2) \\ y(t_2) = 40\sin(t_2) \\ z(t_2) = 0 \end{cases} \tag{24}$$

where $t_2 \in [\pi : -\pi/400 : -\pi]$, it assumes that the curve is constituted of 800 points $[(15, 0, 0), (15.002, 0.31, 0), \cdots, (15, 0, 0)]$, the stat tracking point is $(0, 0, 0)$. The initial position, initial angle, initial velocity, and expected velocity are set as $(x, y, z) = (5, -5, 0)$ m, $(\varphi, \theta, \phi) = (0°, 0, 0)$, $(u, v, w) = (0, 0, 0)$ [kn]. $(u, v, w) = (0.2, 0, 0)$ [kn] for AUV respectively. If the thruster is enwound with foreign matter. Based on the results of thruster fault diagnostics, Figure 12 gives the path tracking results for the circle path. Figure 13 shows the position error for the circle path tracking. The tracking path length is 261.73 m, the average position errors is 0.59 m, the range is $[-10\,\text{m}, 0.71\,\text{m}]$ for AUV tracking position errors, the standard deviation of the tracking error is 0.15 m. The proposed algorithm can realize the circle path tracking with thruster enwinding effectively.

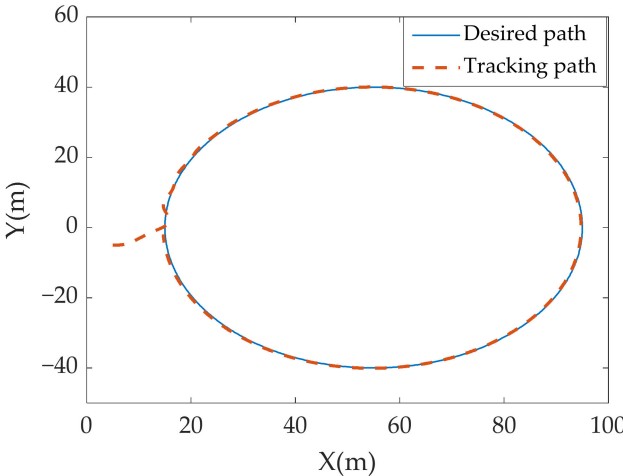

**Figure 12.** Tracking for circle path.

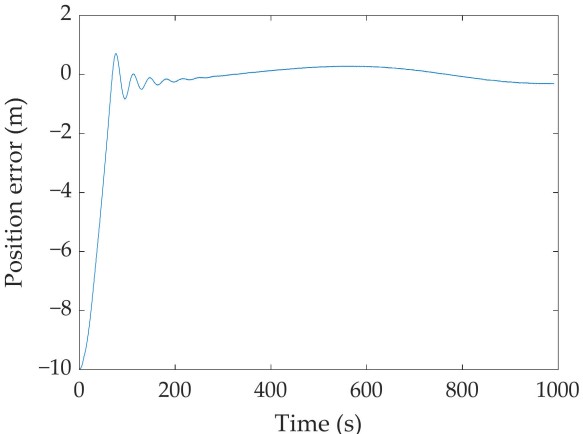

**Figure 13.** Position error for the circle path tracking.

In the real working environment, in order to successfully complete the missions, AUV needs to run along different paths, among which rectangular paths are common. Therefore, a rectangular path is given to simulate the actual path tracking, the basic path parameters are given for the rectangular path as follows:

$$\begin{cases} x = t_3 \\ y = t_3 + 10 & t_3 \in [10, 50] \\ y = -t_3 + 110 & t_3 \in [50, -30] \\ y = t_3 + 170 & t_3 \in [-30, -70] \\ y = -t_3 + 30 & t_3 \in [-70, 10] \end{cases} \tag{25}$$

It assumes that the path is constituted of the points $[(10, 20, 0), (1, 11, 0), \cdots,$ $(10, 20, 0)]$. The start tracking point is $(10, 20, 0)$, The initial position, initial angle, initial velocity, and expected velocity are set as $(x, y, z) = (0, 0, 0)$ m, $(\varphi, \theta, \phi) = (0°, 0°, 0°)$, $(u, v, w) = (0, 0, 0)$ [kn], $(u, v, w) = (0.2, 0, 0)$ [kn]. If the thruster is enwound with foreign matter. Based on the results of thruster fault diagnostics, Figure 14 gives the tracking results for the rectangular path. Figure 15 shows the position error for the rectangular path tracking. The tracking path length is 255.06 m, the average position errors is 0.81 m, the range is $[-1.94$ m, 8.26 m] for AUV tracking position errors, the standard deviation of the tracking error is 0.35 m. The proposed algorithm can realize the rectangular path tracking with thruster enwinding effectively.

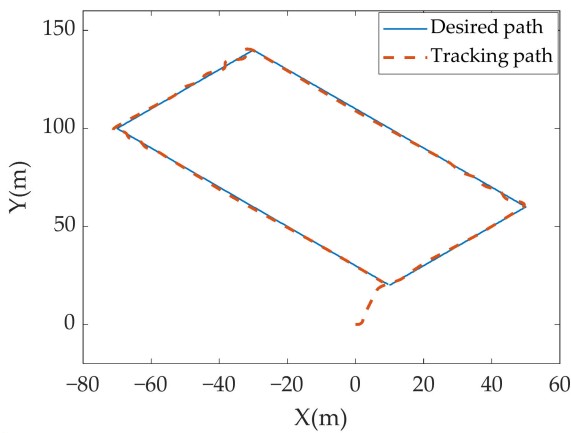

**Figure 14.** Tracking for the rectangular path.

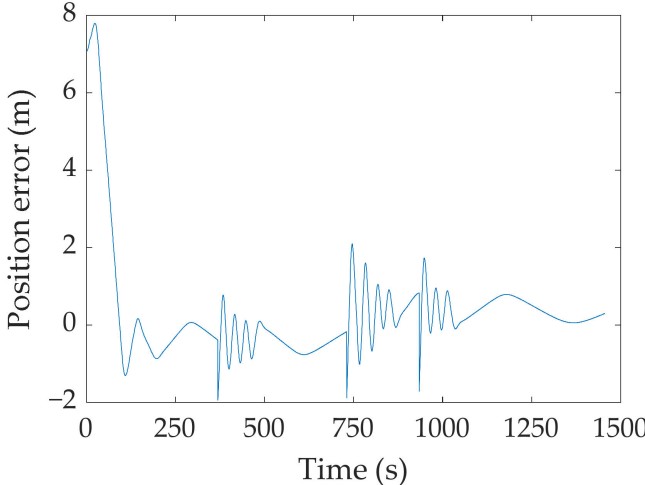

**Figure 15.** Position error for the rectangular path tracking.

In order to verify the path tracking effect of the controller proposed in the presence of obstacles, multiple circular obstacles are set in the environment. The tracking start point is $(x, y, z) = (0, 0, 0)$, the target point is $(x, y, z) = (100, 100, 0)$. The initial position, initial angle, initial velocity, and expected velocity are set as $(\varphi, \theta, \phi) = (0°, 0°, 0°)$, $(u, v, w) = (0, 0, 0)$ [kn]. $(u, v, w) = (0.2, 0, 0)$ [kn]. If the thruster is enwound with foreign matter, the ocean current is 0.07 [kn]. Based on the results of thruster fault diagnostics, it can obtain the fault tolerant control results by the proposed and existing traditional algorithms as shown in Figure 16. For the proposed algorithm, the tracking path length is 150.54 m, the average of the position errors is 0.78 m, and the standard deviation of the position errors is 0.18 m. For the traditional fuzzy control, the parameters of the membership function are optimized by the trial and error method. The tracking path length is 152.71 m, the average of the position errors is 0.92 m, the standard deviation of the position errors is 0.22 m. Table 2 gives the comparison results between the proposed and traditional algorithms. Figure 17 shows the position error for the function tracking. The ranges of the position errors are $[-1.65\ \text{m}, 7.53\ \text{m}]$ and $[-1.75\ \text{m}, 7.47\ \text{m}]$ for the proposed and existing algorithms respectively. Compared with the traditional algorithm, one can see that the tracking path length, average position error, and time are smaller by the proposed algorithm. The proposed algorithm can realize the path tracking in the environment with obstacles and ocean currents effectively.

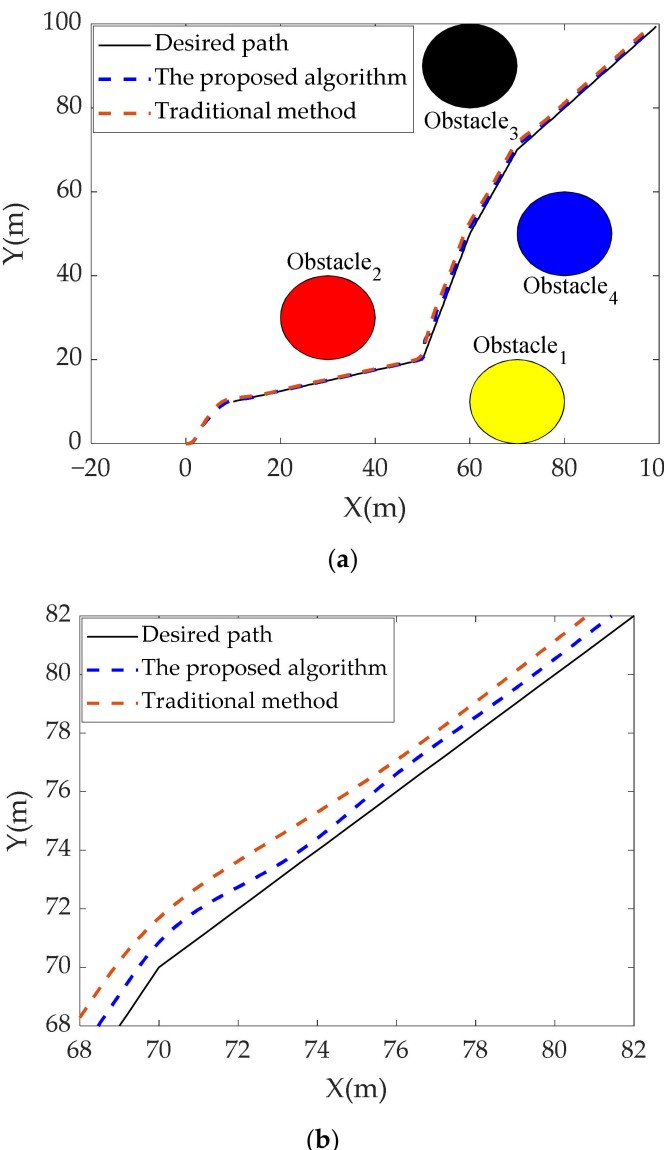

(**a**)

(**b**)

**Figure 16.** (**a**) Path tracking in the environment with obstacles. (**b**) Part of the enlarged view for path tracking curve.

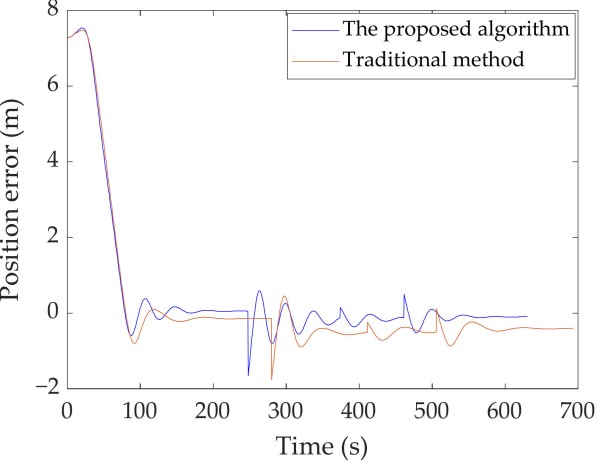

**Figure 17.** Position error for the path tracking in the environment with obstacles.

**Table 2.** Comparison results between the proposed and traditional algorithms.

| Method | Path Length (m) | Mean Tracking Error (m) | Standard Deviation (m) |
|---|---|---|---|
| The proposed algorithm | 150.54 | 0.78 | 0.18 |
| Traditional method | 152.71 | 0.92 | 0.22 |

## 5. Conclusions

Thruster is the driving mechanism for AUV movement, whose fault diagnostics and fault tolerant control are the premise to complete the underwater missions. In practice, ocean currents perturbations could produce noise for thruster fault diagnosis, in this paper, the PFCM algorithm is proposed to solve the problem of thruster fault diagnostics. It is not enough just to realize the thruster fault diagnostics, in order to successfully complete the missions with thruster fault, a fuzzy controller is presented. Considering the effect of ocean currents, the CPSO algorithm is developed to optimize the fuzzy controller, which guarantees the fault tolerant control performance. Based on the designed AUV, a date set is obtained to demonstrate the effectiveness of the thruster fault diagnostics. Different scenarios of path tracking are given to illustrate the performance of the proposed algorithm. Compared with the traditional fuzzy fault tolerant control, the tracking path length and tracking error are smaller by the proposed algorithm, which illustrates the proposed algorithm. In this paper, the proposed algorithm is difficult to be used for weak faults diagnosis of AUV thrusters. However, major faults are generally developed from weak faults. Therefore, in future work, we will try to solve the problem of accurate weak faults diagnosis of AUV thrusters in the presence of interference, which is one of the keys to preventing and reducing catastrophic accidents.

**Author Contributions:** Conceptualization, Q.T. and T.W.; methodology, Q.T.; software, T.W.; validation, Q.T., G.R. and B.L.; formal analysis, Q.T.; investigation, Q.T. and T.W.; resources, G.R.; data curation, T.W.; writing—original draft preparation, Q.T.; writing—review and editing, G.R.; visualization, T.W.; supervision, B.L.; project administration, B.L.; funding acquisition, B.L. All authors have read and agreed to the published version of the manuscript.

**Funding:** This research was funded by the China Postdoctoral Science Foundation (2022M710934), Postdoctoral Applied Research Project of Qingdao City, Project of Shandong Province Higher Educational Young Innovative Talent Introduction and Cultivation Team (Intelligent Transportation Team of Offshore Products).

**Institutional Review Board Statement:** Not applicable.

**Informed Consent Statement:** Not applicable.

**Data Availability Statement:** Not applicable.

**Conflicts of Interest:** The authors declare no conflict of interest.

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
