# Peer review of "Thruster Fault Diagnostics and Fault Tolerant Control for Autonomous Underwater Vehicle with Ocean Currents"

_machines, doi:10.3390/machines10070582_

Round 1

Reviewer 1 Report

This manuscript proposed possibilistic fuzzy C-means (PFCM) algorithm to 16 realize the fault classification for dealing with the effects of the noise and outliers caused by the 17 ocean currents. The topics is very interesting but there are some improvements that need to be done to make this manuscript publishable.

1. Is there any benchmarks/algorithm that has been compared with the proposed method to show that the proposed method is the best method to solve the mentioned problem? If not, please compare with other algorithms/methods to ensure that the implementation of PFCM and CPSO is the best to solve the problem. It seems that there is not comparison with other algorithms as Figure 11 until 18 only shown the result from the proposed algorithm. Author should compare with other works for verification purpose.

2. Label the parts of the AUV in Figure 8. It is suggested that if author can show the image of the designed AUV closer and label the parts of AUV.

Reviewer 2 Report

See the attached review file

Author Response

    The article suits the scope of Machines. The subject refers to a permanent concern and challenge for research & development in the field of autonomous underwater vehicles (AUV), namely thruster fault diagnostics and fault tolerant control. Such a theme is worthy of investigation, being one of great interest for the scientific community in the field. In this regard, a possibilistic fuzzy C-means (PFCM) algorithm is proposed to deal with the effects of the noise and outliers caused by the ocean currents. Simulation and underwater experiments are used to verify the accuracy and feasibility of the proposed method. Several revisions are necessary for increasing the scientific value of the work and clarifying the approach, as indicated below.

1. The paper should be written in an impersonal way (but this also depends on the editorial policy).

Response

    In line with the comments, we have rewritten the paper in an impersonal way.

2. In the introductory section, the main objectives of the work should be more clearly formulated in response to the critical analysis of the current research stage, going through the next path: the disadvantages of the existing methods, the research opportunity, the innovative approach.

Response

    In line with the comments, we have revised the introduction section of the paper. The current studies have given some methods to solve the AUV fault diagnosis. However, ocean currents perturbations could produce noise for thruster fault diagnosis, the current methods are difficult to be used for AUV fault diagnosis with ocean currents in practice effectively. The above methods also not consider how to control AUV to complete the underwater missions with minor faults. We have added the description in the introduction section. Then, we given the description of our proposed innovative approach in the paper.

3. In the background research, the authors should refer to their previous works in the field (if any), and highlight the elements that differentiate the approaches.

Response

    Thank you very much for your suggestions. Although we have done some work for fault detection for nonlinear systems, we haven’t done the previous works in the field of AUV fault diagnostics.

4. The research methodology should be more clearly described.

Response

    We have added the descriptions in the text to make the paper clearer, which includes the further explanation of PFCM and CPSO algorithms in this paper.

5. The 2nd section of the paper shows very little information, not being justified as a separate chapter.

Response

    In line with the comments, we have moved the 2nd section in to the next section.

6. All parameters used in equations and figures should be explained with the first use; carefully check the whole paper in this regard.

Response

    Thank you very much for your comments and suggestions. All parameters used in equations and figures are explained in the first use in this paper, we have checked the whole paper.

7. There is a discontinuity in the notation of the figures (figure 4 is missing)

Response

    In line with the comments, we have change “figure 5” into “figure 4” in the paper.

8. Check the references to the figures on page 9, lines 292-293.

Response

    In line with the comments, we have rearranged the figures in order.

9. The conclusions section should be extended & improved by a more detailed discussion on the research findings as well as on the future research directions, in terms of research opportunities opened by this work.

Response

    We have added more detailed discussion on the research findings. Compared with the traditional fuzzy fault tolerant control, the tracking path length and tracking error are smaller by the proposed algorithm. In this paper, the proposed algorithm is difficult to be used for weak faults diagnosis of AUV thrusters. However, major faults are generally developed from weak faults in fact. Therefore, in the future work, we will try to solve the problem of accurate weak faults diagnosis of AUV thrusters in the presence of interference, which is one of the keys to prevent and reduce catastrophic accidents.

Reviewer 3 Report

This work presents a thruster fault diagnostic (a possibilistic fuzzy C-means algorithm), to realize the fault classification for dealing with the effects the ocean currents. A fuzzy control strategy is proposed to solve the fault tolerant control for AUV. Simulation and underwater experiments are used to verify the accuracy and feasibility of the proposed.

The provided information is relevant for the knowledge field. Nevertheless, some issues should be addressed before this manuscript could be considered for publication.

1) Ocean currents perturbations could produce noise for thruster fault diagnosis. Nevertheless, it is not clear to say that: “effects of the noise and outliers caused by the ocean currents”. This comment should be clarified.

2) Figure 3. Should be mentioned in the text.

3) Figure 4. Is not presented.

4) Figure 7. Should be mentioned in the text.

5) Mentions (in the text), to figures 4 and 5, corresponds to figures 8 and 9.

6) In section 5 “Simulation and experiment analysis”, detailed implementation information should be provided (hardware, software, configuration, settings).

7) The Conclusion section should include quantitative results, advantages and disadvantages, limitations, and recommendation for real implementations.

Author Response

     This work presents a thruster fault diagnostic (a possibilistic fuzzy C-means algorithm), to realize the fault classification for dealing with the effects the ocean currents. A fuzzy control strategy is proposed to solve the fault tolerant control for AUV. Simulation and underwater experiments are used to verify the accuracy and feasibility of the proposed. The provided information is relevant for the knowledge field. Nevertheless, some issues should be addressed before this manuscript could be considered for publication.

1) Ocean currents perturbations could produce noise for thruster fault diagnosis. Nevertheless, it is not clear to say that: “effects of the noise and outliers caused by the ocean currents”. This comment should be clarified.

Response

    In line with the comments, we have clarified the corresponding comment with the statement of “Ocean currents perturbations could produce noise for thruster fault diagnosis”.

2) Figure 3. Should be mentioned in the text.

Response

    In line with the comments, we have mentioned figure 3 in the text.

3) Figure 4. Is not presented.

Response

    We have change “figure 5” into “figure 4” in the paper.

4) Figure 7. Should be mentioned in the text.

Response

    We have mentioned the figure in the text.

5) Mentions (in the text), to figures 4 and 5, corresponds to figures 8 and 9.

Response

    In line with the comments, we have rearranged the figures in order.

6) In section 5 “Simulation and experiment analysis”, detailed implementation information should be provided (hardware, software, configuration, settings).

Response

    The algorithm is coded in MATLAB R2019a and simulations are run on the PC with 2.00 GHz CPU/8 GB RAM. We have added the description in the paper.

7) The Conclusion section should include quantitative results, advantages and disadvantages, limitations, and recommendation for real implementations.

Response

    Compared with the traditional fuzzy fault tolerant control, the tracking path length and tracking error are smaller by the proposed algorithm. The advantages of the proposed algorithm are that it can deal with the thruster fault diagnostics and fault tolerant control effectively. In this paper, the proposed algorithm is difficult to be used for weak faults diagnosis of AUV thrusters. However, major faults are generally developed from weak faults in fact. Therefore, in the future work, we will focus on to solve the problem of accurate weak faults diagnosis of AUV thrusters, which is one of the keys to prevent and reduce catastrophic accidents.

Round 2

Reviewer 3 Report

The authors addressed the recommendations, the manuscript has been sufficiently improved and could be considered for publication.